# Public expectations about the impact of COVID-19 on climate action by citizens and government

**Ivan Savin**[1,2]*, **Stefan Drews**[1], **Jeroen van den Bergh**[1,3,4], **Sergio Villamayor-Tomas**[1]

**1** Institute of Environmental Science and Technology, Universitat Autònoma de Barcelona, Cerdanyola del Valles, Barcelona, Spain, **2** Graduate School of Economics and Management, Ural Federal University, Yekaterinburg, Sverdlovsk Region, Russian Federation, **3** ICREA, Barcelona, Spain, **4** School of Business and Economics & Institute for Environmental Studies, VU University Amsterdam, Amsterdam, The Netherlands

* ivan.savin@uab.cat

**Data Availability Statement:** All the relevant data and the code to reproduce the results are available at the following link at GitHub https://github.com/IvanVSavin/PLOS_One_PublicExpectationsCOVID.git.

## Abstract

Since the onset of the COVID-19 crisis many have opinionated on how it may affect society's response to climate change. Two key questions here are how COVID-19 is expected to influence climate action by citizens and by the government. We answer these by applying topic modelling to textual responses from a survey of Spanish citizens. The identified topics tend to be more negative than positive, and more optimistic concerning future climate action by citizens. Positive views involve increasing pro-environmental behavior and are more common among younger, higher educated and male respondents as well as among those who perceive climate change as a serious threat or positively assessed COVID-19 confinement. Negative topics express concern that financial resources for climate action will be limited due to a focus on healthcare and economic recovery. In addition, they mention government mismanagement and waste due to use of protective measures like masks and gloves as impediments to effective climate action.

## Introduction

Speculations about how COVID-19 will affect climate action abound. Optimistic voices express that it has already considerably changed individual behaviors in an environmentally favorable direction; that governmental responses to COVID-19 will encourage such action and thus accelerate a low-carbon transition [1, 2]; or that it can increase support for stringent climate policies [3]. More critical voices have pointed out that environmental concerns are of low priority in the economic recovery programs, such as in China [4]. It is also possible that most people will simply revert to their previous routines. Private transport may even grow as many people consider public transport as too risky in terms of virus contagion [5].

The above and other perspectives are not just voiced in academic but also in public debates. Knowing about these public opinions can help to derive insights about people's consumption and voting behaviors, which form the basis for private and public climate action. Knowledge about people's expectations is also useful for policy makers when adapting climate policy both

**Funding:** This work was funded by an ERC Advanced Grant from the European Research Council (ERC) under the European Union's Horizon 2020 Research and Innovation Programme [grant agreement n° 741087] and by a María de Maeztu Program for Units of Excellence awarded to ICTA-UAB by the Spanish Ministry of Science and Innovation [grant number CEX2019-0940-M]. I.S. acknowledges financial support from the Russian Science Foundation [RSF grant number 19-18-00262]. S.V.-T. was supported by the Ramón y Cajal Fellowship (Ministerio de Ciencia y Universidades, nr. RyC-2017-22782).

**Competing interests:** The authors have declared that no competing interests exist.

during and after the pandemic. European governments have shown increasing interest in integrating citizen inputs into their foresights and agendas for both legitimacy and innovation reasons. This ambition, however, has its own challenges, including most prominently the difficulties of synthesizing large qualitative data meaningfully [6]. Our study addresses this challenge and contributes to previous literature on participatory foresight and scenario planning based on open-ended public-opinion surveys [7]. Specifically, it applies structural topic modelling to a large, country-wide sample of open-ended survey data.

To this end, we undertook a large nationally representative survey from Spain. Spain was among the first countries being severely hit by the pandemic and since then continued to be among those most severely affected, both health-wise and economically [8, 9]. By the 20th of March 2020, Spain reached a peak of more than 10,000 daily cases and by the 1st of April the difference between current deaths and expected deaths (as per the previous year) attained a maximum value of 1848 (see S1 Fig in S1 File for more details). In line with this, the Spanish government implemented one of the strictest lockdowns in Europe from mid-March (14th of March) to mid-June (21st of June). All this makes Spain an excellent case to explore people's expectations on how COVID-19 will affect citizens' and governmental climate action [10, 11].

We use the method of structural topic modelling (STM) to analyze responses to two open-ended survey questions on the perceived effects of COVID-19 on future personal and governmental climate action. Using open-ended, rather than closed-ended, survey questions has the advantage of generating a wide range of insights. This is particularly helpful since little prior knowledge is available on this topic. In addition, we examine how the generated topics are linked to various respondent characteristics, as well as perceptions of, and experiences with, COVID-19 and climate change. This allows to understand not just the content of people's expectations, but also who tends to express optimism or pessimism regarding future climate action. In doing so, we simultaneously apply qualitative and quantitative research strategies: the first aim to identify the "internal systemic understanding of social realities", while the second tests it by establishing a relation between topics and different socio-economic groups of respondents.

STM is a method to classify textual responses into distinct topics developed specifically for open-ended survey responses [12]. It has already been used in about 60 articles on a wide range of topics [13], including those published on public perceptions of climate change and air pollution [14–16], associations with carbon taxation and its fairness perception [17], people's beliefs about others' climate beliefs [18] classification of startup companies and identification of their global trends [19], and studies published on environmental innovation and societal transitions [20]. Applications of topic modelling have been even broader including, for example, tracing development of agricultural and water technologies over long time [21] and understanding public attitudes towards municipal solid waste sorting policy in China [22]. For a recent literature review on applications of topic modelling see [23].

## Materials and methods

### Data collection

In this study we draw on two open-ended questions from an online survey conducted in June-July 2020 in Spain. The sample includes 2200 participants and is representative of the general population in terms of age, gender and geographical distribution. Furthermore, comparison of the distribution of political orientation in our survey with that of the Spanish Centre for Sociological Studies from June 2020 shows no statistical difference (Mean (M) = 4.44, Standard Deviation (SD) = 2.41 versus M = 4.6, SD = 2.0). This is important as political views tend to strongly correlate with attitudes to climate change and policy [24, 25]. Respondents from our

survey, however, tend to be more educated and have a higher income (see S2 Table in S1 File). Thus, similar to most other online surveys, our sample is underrepresented with people having a low education or income. The survey contained about 40 questions and participants took on average 19 minutes to complete it. The response rate was 68%.

The survey was approved by the Ethics Committee on Animal and Human Experimenta-tion (CEEAH for its Spanish-language acronym; reference number of the case: 5226) of the Autonomous University of Barcelona. The survey was conducted in Spanish language by a professional survey company "Netquest". The survey company Netquest hosts a large panel of about 1.3 million people living in 23 countries. Data collection was conducted in accordance with the general Market Research ISO norm 25252 using "own software developments for online surveys" (https://www.netquest.com/en/online-surveys-investigation). If people want to be part of the panel in order to respond to questionnaires, they have to previously accept the explicit informed consent. If people want to be part of the panel in order to respond question-naires, they have to previously accept the explicit informed consent. The explicit informed consent is provided by the panelists through the acceptance of the checkbox available on the registration page (see https://www.nicequest.com/es/legal), corresponding to the conditions of use of the panel and its privacy policy, respectively. It is not possible to register on the panel without prior acceptance of that legal information. The informed consent then includes aspects such as that the company only use respondents' data to compile statistics; that respon-dents can leave the panel at any time by accessing their personal Nicequest account and if they do so, they will lose the conches they have accumulated which can be later exchanged for prod-ucts and services. The company also informs that they can also eliminate respondents account if they do not collaborate honestly.The analysis was undertaken using the text in the original language, to avoid possible inconsistencies due to translation, while the results of this are reported in English (some results in Spanish are provided in the S1 File).

The two open-ended questions were posed early on in the survey to avoid potential influ-ence from other questions on the responses. The precise procedure was as follows. At the beginning of the survey, respondents were asked about the main problems that exist in Spain, allowing them to select from a list of more than 10 options (e.g., climate change, spread of infectious diseases, terrorism). The second question was: "Two issues that receive a lot of atten-tion nowadays are climate change and the COVID-19 (Coronavirus) pandemic. Overall, how do you think the COVID-19 pandemic will affect actions of the government towards climate change?". In a closed-ended response format, participants could evaluate the effects of COVID-19, using a 5-point Likert scale from (1) "very negatively" to (5) "very positively". To get an idea what thoughts have led people to answer that way, we then posed the first open-ended question asking them to explain their views:

> "Can you explain in your own words why you think that Covid will affect [answer that was chosen by participants] the actions of the Spanish government with regard to climate change? We would like that you take your time to answer this question and write some sen-tences. All kinds of answers are welcome."

Subsequently, a control question was posed to identify respondents who were not paying close attention to questions. In addition, we asked respondents whether they live in urban, peri-urban or rural area as this may related to the carbon intensity to their lifestyle and their opinions on cli-mate action and policy. These two questions also served to distract respondents from their responses to the first set of COVID-19 questions and thus minimize inference with these. In the following, again a closed question preceding its open-ended counterpart was posed: "Overall, how do you think the COVID-19 pandemic will affect actions of Spanish citizens towards climate

change?" asking participants to choose between "very negatively", "negatively", "neutrally", "positively", "very positively". The second open-ended question then was phrased as:

> "Can you explain in your own words why you think that Covid will affect [answer that was chosen by participants] the actions of the Spanish citizens with regard to climate change? We would like that you take your time to answer this question and write some sentences. All kinds of answers are welcome."

Combining the closed- and open-ended questions allows us to have both, an overall evaluation of the expectations (from very negative to very positive) and the textual explanation for these expectations. We exploit this feature when evaluating the topics as either positive or negative. The questions were formulated as general as possible. This allows participants to freely express their expectations about the subject, not limiting them by any additional information. This approach is in line with previous studies [14, 26]. As we show below, posing the questions this way permits to uncover many different concerns of the respondents.

The average (median) length of responses to the first open question is 19 (13) words, while for the second 16 (12) words. In both questions the shortest response was one word. S2 Fig in S1 File demonstrates the complete distribution of length of answers for both questions.

## Data analysis

Before proceeding with applying STM to the open responses, the data was pre-processed. This involved:

- Replacing capital by lower-case letters;

- transforming words to their dictionary form by lemmatization;

- retrieving frequent expressions (so called bigrams, like "climate_change", "renewable_-energy") using Normalized Pointwise Mutual Information score (NPMI, [27]);

- deleting stop words (like pronouns and prepositions and other common words);

- removing words with a length of less than three characters or appearing less than four times in all answers.

This resulted for the first open-ended question in 24 discarded answers, while the remaining responses contain 701 (unique) words and 16092 total words. For the second open-ended question 28 responses were discarded, while the remaining responses contain 612 (unique) words and 14381 total words.

To classify the responses into meaningful topics, we utilize the structural topic modelling method (see [17] for a detailed discussion). In short, TM makes a Bayesian inference of words related to a given topic and the topics being discussed in each survey response, based on responses already observed. TM assumes that each word in the responses is generated through a two-step process: first, each response has its own distribution of topics, and a topic is randomly drawn from it; second, each topic has its own word distribution, and a word is randomly drawn from this distribution for the topic selected in the first step. Essentially, each description is a result of repeating these two steps many times. Therefore, the responses may have multiple topics present in them in different proportions. Topic modelling discovers the topic distribution for each survey response and the word distribution of each topic iteratively, by fitting this two-step procedure to the observed descriptions until it finds the best model that describes the underlying data.

The advantage of this method over a simple word count is that words are considered not in isolation but in the context of other words they appear together with. For example, observing

words "dedicate" and "money" next to "crisis" in the topic with label "Insufficient resources to address climate change" (see S7 Fig in S1 File) suggests that people expressed the need to spend the money to fight the crisis. An advantage of STM compared to other topic modelling algorithms is that it was developed particularly for short texts typical for survey responses to open questions. By incorporating additional information about the responses, such as the respondents' beliefs, age or acceptability of the carbon tax, STM produces higher quality topics [12]. STM has been implemented using the associated package in R developed by [13].

As additional information to form meaningful topics, we use several covariates obtained from the survey for both open questions (see S1 Table in S1 File). One set of variables concerns socio-demographic characteristics, namely gender, age, education. Regarding age and gender, for example [28], demonstrated important differences among people in terms of knowledge about and behavior responding to the risk of being infected by COVID-19. Two further, comparable variables captured people's perceived threats from climate change and COVID-19, respectively. Past research shows that risk perception is an important predictor of climate- and health-related behaviors [29, 30]. Two other variables explore COVID-19-related consequences and experiences, namely the effect of the COVID-19 crisis on household income, and the overall perceived experience with COVID-19 confinement. One variable covers climate policy attitudes in the form of individuals' acceptability of a carbon tax, which complements climate risk perceptions noted before. In addition, for the first open-ended question we included respondents' evaluation of the government's role in fighting COVID-19 (a type of collective efficacy belief), as well as the closed-ended question on beliefs about the positive or negative impact of COVID-19 on the government's climate policies, which preceded the open question. For the second open-ended question, we use similar covariates, with the difference that the final two covariates (beliefs on collective efficacy and effect of COVID-19 on climate policy) are directed towards people instead of the government (S3 Fig in S1 File). These variables were chosen in a rather exploratory manner given the lack of literature which could explain variation in public responses on these issues. From S3 Fig in S1 File, showing the distribution of answers, we can see that people are more pessimistic in evaluating how COVID-19 will affect climate action by people than by the government. Furthermore, respondents are more benevolent in evaluating people than the government regarding fighting COVID-19.

As we demonstrate in S4 Fig in S1 File, we did not find strong correlations between any pair of covariates included in each of the two topic models. The only exception was views on how COVID-19 affected actions by the government and people with regard to climate change, but these are included in different models. The Variance Inflation Factor varies in our models between 1.048 and 1.276 (much lower than the benchmark of 5) suggesting absence of multicollinearity in our regression analysis.

Before proceeding with results, we have to decide upon the number of topics k. To this end, we run the STM algorithm for different number of topics between three and twenty and evaluate the results on three criteria. First, how well the resulting models predicts the data on a sample held-out from the estimation step ('heldout log-likelihood'). Second, how much the most frequent words in the generated topics coincide ('exclusivity'). Third, whether words from the same topic tend to appear in the same responses ('semantic coherence'). A fourth implicit criterion is whether the model is easy to overview and interpret, which favors fewer topics. The top panel in S5 Fig in S1 File summarizes the results for the first open-ended question, while the lower panel of that figure does the same for the second open question. In the first case we arrived at 13 topics, and in the second at 12. Increasing the number of topics further would not considerably increase the prediction accuracy and exclusivity of the associated models, while it would increase their complexity.

## Results and discussion

### Respondents' associations with governmental climate action

Table 1 summarizes the 13 topics identified from the first open question about how COVID-19 will affect climate action by the government. It shows the 10 most discriminating (frequent and exclusive) words for each topic together with an illustrative response having a large prevalence (i.e. weight) of that topic along with the share of the overall set of responses belonging to each of the topics (topic proportion). After exploring many more illustrative responses, which we do not include here for brevity reasons, we decided on topic labels that reflect their main themes in a clear and concise way.

**Table 1. Topics identified for responses to the first question on governmental climate action.**

| | Topic label | Most discriminating terms and illustrative responses | Topic proportion |
|---|---|---|---|
| 1T1 | Insufficient resources for climate | resource, money, crisis, invest, investment, lack, dedicate, enough, require, economic | 8.0% |
| | | "There will not be sufficient resources due to the great economic cost of the crisis" | |
| 1T2 | COVID-19 and climate change are independent | believe, affect, follow, change, equal, different, although, thing, influence, modify | 8.2% |
| | | "I don't really think one thing affects the other" | |
| 1T3 | Priority for COVID-19 | attention, background, related, action, climate_change, leave, lend, problem, covid, priority | 12.5% |
| | | "Because they are going to leave behind all the actions regarding climate change to focus on the covid" | |
| 1T4 | Government responds inadequately to COVID-19 | do, relationship, spanish, affair, none, capable, disaster, manage, respect, correctly | 5.3% |
| | | "Simply because the Spanish government does not have the slightest idea of acting correctly in these cases so Spain is going to go into a very severe crisis" | |
| 1T5 | COVID-19 distracts from climate | now, virus, pandemic, importance, currently, never, important, take_advantage, consider, first | 5.6% |
| | | "I don't think climate change is taken into account while there is another distraction on the table they have never given it the importance it deserves and obviously with an ongoing pandemic they give it even less importance" | |
| 1T6 | Self-interested politicians | know, situation, think, pass, worry, come, manage, import, same, only | 6.0% |
| | | "Because politicians are only interested in their pockets, they are not doing anything for anything or for anyone except themselves" | |
| 1T7 | COVID-19 adds to waste | more, mask, plastic, glove, residue, put, material, disposable, throw_away, look | 8.1% |
| | | "Use of gloves and masks are disposable and they generate a lot of waste" | |
| 1T8 | Teleworking and less travel | environment, transport, major, polluting, consumption, negative, vehicle, energy, promote, private | 6.5% |
| | | "The use of alternative transport means such as bicycles and teleworking will be incentivized in such a way that the number of trips to job places will be reduced" | |
| 1T9 | Priority for economy and health | economy, politics, priority, country, have, health, environmental, clear, give_priority, reactivate | 7.9% |
| | | "The government will have to value that employment, the economy and health are priorities compared to climate change" | |
| 1T10 | Spend climate money on COVID-19 | fight, spending, combat, effort, prioritize, fund, center, destined, allocate, expenditure | 5.1% |
| | | "Because part of the effort and financial means allocated to the fight against climate change will be used to fight the covid" | |
| 1T11 | Government makes bad decisions | politician, bad, a_bit, decision, turn, totally, majority, much, benefit, unique | 8.2% |
| | | "Because until now the government has not cared about placing the best prepared people to solve the problems that arise in Spain, they have only cared about placing their friends" | |
| 1T12 | People stay at home and pollute less | take, measure, pollution, keep, awareness, wait, adopt, see, decrease, car | 8.9% |
| | | "Thanks to the covid and the quarantine, people did not leave the house, they did not move by car, thus reducing pollution" | |
| 1T13 | COVID-19 is environmental wake-up call | confinement, bill, planet, nature, positive, activity, human, return, stop, have | 9.7% |
| | | "They have noticed the improvement of the planet during the confinement" | |

Note: The terms shown are those that are the most frequent as well as exclusive to each topic. Illustrative responses are chosen from the ten responses with the highest topic prevalence. The original text was in Spanish. Here we report only the English translation, while the original Spanish words and phrases are presented in S3 Table in S1 File.

S6 Fig in S1 File presents word clouds illustrating dominant terms for the resulting topics, with font standing for likelihood of the words and darkness for exclusivity. For example, in 1T5, suggesting that COVID-19 is a distraction from climate change, the words "pandemic" and "now" come out very strongly, which suggests the urgency to deal with the COVID-19 crisis and postpone other issues (henceforth, 1TX and 2TX denote topic X from the first and second open-ended question, respectively).

As one can see, some topics reflect criticism of the government (1T4, 1T11), politicians in general (1T6), or the concern about budgetary deficit and the resulting reduction of spending on climate-related policies (1T1, 1T9, 1T10). Other topics concentrate on the trade-off between climate and COVID-19 policies (1T3, 1T5) and the fact that people could reduce pollution in the time of lockdown (1T8, 1T12). The reason why some topics have similar meanings but are recognized as separate clusters of words is that people used different words to express their thoughts. More unique topics in terms of their meaning are 1T2 (denial that the two issues are related), 1T7 (increasing pollution because of disposable protection equipment) and 1T13 (increasing awareness about environmental problems).

S8 Fig in S1 File presents correlations between topic prevalence across responses. These allow one to see if any pair of topics was mentioned very often or very rarely together. As one can see, topics 1T4, 1T6 and 1T11 criticizing the government or politicians, and 1T3 and 1T10 about focusing attention and resources are indeed strongly correlated. Other strong pairs are 1T12 and 1T13 (people staying at home and polluting less become more aware about the consequence of their actions) and 1T2 and 1T4 (expecting no mutual effect and criticizing the government).

We proceed with a regression analysis quantifying factors explaining the variation of topic prevalences among the responses. As explained in the Methods section, we selected ten covariates to form the STM topic model and now use them here in a linear regression model specified for each of the 13 topics (indexed with k) as demonstrated below:

$$
\begin{aligned}
\text{Topic Prevalence}_k \sim {}& \text{Constant}_k + \text{Age} + \text{Gender} + \text{Education} + \text{Perceived threat from climate change} + \\
& \text{Perceived threat from COVID}-19 + \text{Overall experience with COVID-19} + \\
& \text{Income change due confinement} + \text{Carbon tax acceptance} + \qquad (1)\\
& \text{Expectations about governmental climate action} + \\
& \text{Evaluation of government fighting COVID}-19 + \text{Residual}_k
\end{aligned}
$$

The full set of results is reported in S5 Table in S1 File, while in Fig 1 only results significant at 1% level and for the first eight out of ten covariates in the regression Eq (1) are given (the last two covariates are addressed separately in the following section). Whenever we observe a prevalence of topic k to significantly differ across a given covariate, this implies that responses from people differing on that covariate also significantly differed in expressing that topic. Given that many covariates we used represent views and perceptions that are interdependent, these results should be understood as a correlation analysis.

For example, men are more likely to stress the problem of lack of resources compared to women (1T1, 1T9), who are more concerned with increasing waste of disposable masks and gloves (1T7). Similarly, more educated respondents talk more about lack of resources which have to be directed to fight against COVID-19 as well as the benefits from teleworking (1T1, 1T8, 1T9), while less educated people tend to criticise more the government and politicians (1T4, 1T6, 1T11). People who support carbon taxation have more negative expectations compared to opponents, as they suggest that more attention and priority will be given to COVID-19 and the economic crisis (1T3, 1T9). Like less educated people, opponents of carbon taxation tend to concentrate on criticising the government and politicians.

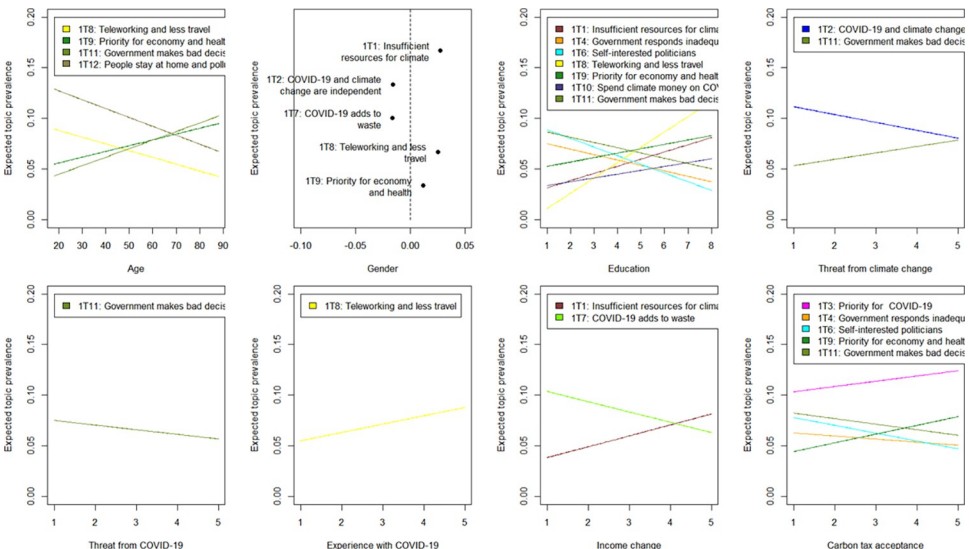

**Fig 1. Effect of covariates on topic prevalence for thoughts on governmental climate action.** Note: Point estimates of selected covariates on topic prevalence are plotted holding all other covariates constant. The plot for gender shows mean difference in topic proportions between male and female (a positive value on the X-axis indicates a larger prevalence for male respondents). Only results significant at least the 1% level are displayed. Results for all topics are available in S5 Table in S1 File.

Compared to the basic socio-demographic characteristics (like age, gender and education), other covariates (such as perceived threats from COVID-19 and climate change, overall personal experience with COVID-19-related confinement and its effects on household income) do not explain much variation in topics people expressed in their open responses.

## Respondents' associations with people's climate action

We now repeat the described steps of analysis for the responses to the second open question concerning which associations respondents have with future climate actions by people after having experienced the COVID-19 pandemic. Table 2 summarizes the results for the 12 topics identified. Word clouds for the resulting topics are presented in S7 Fig in S1 File. Again, in some topics few words come out very strongly in terms of frequency and exclusivity (like "citizen", "do", "same" in topic 2 or "government", "must", "help" in topic 4) further clarifying their main message.

As we can see, several topics appear similar to topics from the first open question. These include 2T9 on increasing waste of disposable masks and gloves and 2T11 on benefits from teleworking, which are akin to 1T7 and 1T8 on governmental climate action, respectively. One way to interpret this is that at least some people did not differentiate sufficiently between the two open questions and gave similar responses. This is supported by S10 Fig in S1 File correlating prevalence between the two sets of topics, though correlations are not as high as one might expect (below 0.5). New thoughts expressed in the remaining topics are an increased public awareness regarding consumption (2T1, 2T12), little effect of COVID-19 on people's behavior (2T2, 2T7), a focus on other urgent problems (2T3, 2T8, 2T10), a rising awareness about environmental problems (2T5, 2T6) and a lack of governmental support (2T4).

S9 Fig in S1 File again helps to identify topic pairs based on the frequency respondents mentioned them together. These are topic pairs 2T1 & 2T11 and 2T5 & 2T12 (people become more aware and behave more responsibly) or 2T2 & 2T8 (people continue acting as before and focus on the present). These results in S8 and S9 Figs in S1 File. thus allow us to consider topics not

**Table 2. Topics identified for responses to the second question on people's climate action.**

| | Topic label | Most discriminating terms and illustrative responses | Topic proportion |
|---|---|---|---|
| 2T1 | More awareness and less consumption | take, go_out, consume, awareness, pollution, displace, even, home, aware, need | 6.4% |
| | | "People are becoming more aware of it, the bicycle is used much more, it is consumed with more awareness" | |
| 2T2 | People act as before | do, follow, same, a_bit, influence, citizen, topic, explain, action, covid | 9.8% |
| | | "They will continue acting the same as they did before covid" | |
| 2T3 | Priority for economy and health | change, affect, economy, relationship, situation, health, habit, environment, economic, thing | 7.1% |
| | | "That we prioritize health or the economy if we decide for health we do not generate money if we prioritize the economy the virus spreads who cares about climate change" | |
| 2T4 | Lack of government support | government, duty, help, put, means, environmental, citizenship, environmental, motive, unemployment | 6.1% |
| | | "Because people are not sufficiently aware of the damage we are causing and it should be governments that force people to act through education" | |
| 2T5 | Confinement fosters environmental care | know, take_care, improve, world, realize, air, have, human, city, great | 9.8% |
| | | "The pandemic has taught us the importance of respecting and caring for the environment and how vulnerable we are" | |
| 2T6 | Change is inevitable | change, want, time, a_lot, carry, new, virus, appearance, suffer, activity | 5.1% |
| | | "People have spent time at home, time to reflect, time in which we realize that we cannot buy what is important with money and anything goes, I think we will learn a lot from this pandemic" | |
| 2T7 | Old habits die hard | think, life, how_much, normality, habit, reason, sense, believe, always, attitude | 7.4% |
| | | "I think in the long run they will forget and return to their routines" | |
| 2T8 | People are myopic | now, equal, important, import, good, concern, attention, pending, resource, protect" | 7.8% |
| | | "People live day by day and they don't care what happens tomorrow as long as they are fine now" | |
| 2T9 | COVID-19 adds to waste | mask, glove, plastic, use, throw_away, residue, soil, generate, street, public | 13.1% |
| | | "The massive use of masks and gloves will end up as garbage in our oceans" | |
| 2T10 | People are occupied by other problems | climate_change, problem, worry, majority, worried, spanish, priority, exist, position, population | 12.0% |
| | | "Because it is another problem and we already have too many problems" | |
| 2T11 | Teleworking and less travel | less, car, respect, travel, catch, count, decrease, trip, see, energy | 9.2% |
| | | "Transportation by car and motorcycle will be reduced and more will be done by bicycle and scooter, teleworking will be promoted avoiding unnecessary travel" | |
| 2T12 | People consume more responsibly | aware, nature, though, awareness, behavior, impact, response, power, reflect, better | 6.2% |
| | | "This forced confinement has made people internalize that there is a transition to a world that is more respectful of our environment and with" | |

Note: The terms shown are those that are the most frequent as well as exclusive to each topic. Illustrative responses are chosen from the ten responses with the highest topic prevalence. The original text was in Spanish. Here we report only the English translation, while the original Spanish words and phrases are presented in S4 Table in S1 File.

just in isolation but also in relation to each other, and hence derive explanations of any positive or negative expectations about future climate action by people and government. To illustrate, people have many problems already (2T10) and therefore focus on the present (2T8).

Fig 2 displays the results of our regression analysis using the following equation:

$$
\begin{aligned}
\text{Topic Prevalence}_k \sim\ & \text{Constant}_k + \text{Age} + \text{Gender} + \text{Education} + \text{Perceived threat from climate change} + \\
& \text{Perceived threat from COVID}-19 + \text{Overall experience with COVID-19} + \\
& \text{Income change due confinement} + \text{Carbon tax acceptance} + \\
& \text{Expectations about people's climate action} + \\
& \text{Evaluation of citizens fighting COVID}-19 + \text{Residual}_k
\end{aligned}
\tag{2}
$$

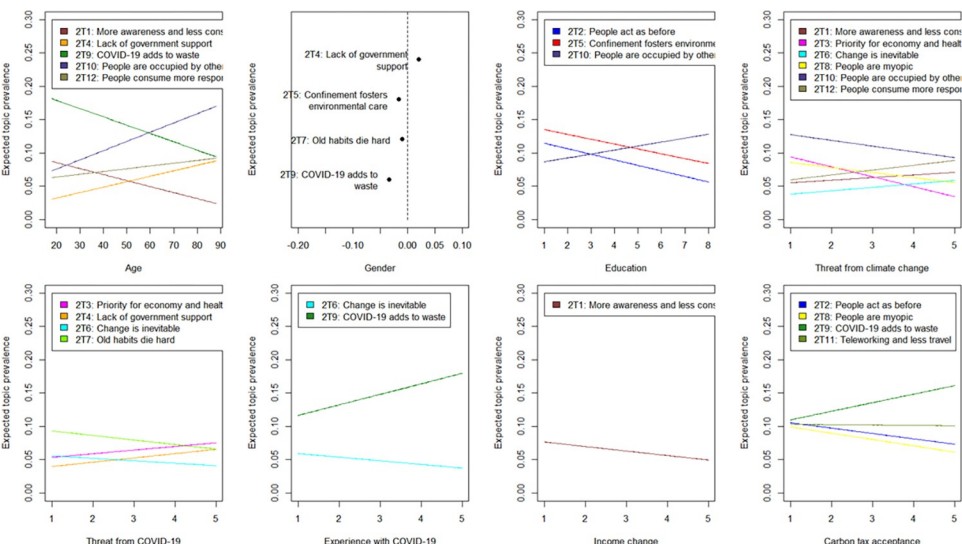

**Fig 2. Effect of covariates on topic prevalence for thoughts on people's climate action.** Note: Point estimates of selected covariates on topic prevalence are plotted holding all other covariates constant. The plot for gender shows mean difference in topic proportions between male and female (a positive value on the X-axis indicates a larger prevalence for male respondents). Only results significant at least the 1% level are displayed. Results for all topics are available in S6 Table in S1 File.

The full set of results is presented in S6 Table in S1 File, while Fig 2 demonstrates results for the first eight listed covariates and only for those topics significant at 1% level. Compared to expected action by the government, here we observe a stronger influence of perceived threats from COVID-19 and climate change and a weaker influence of education in explaining variation of identified topics among people. Men are more likely to stress the lack of support from the government in combating climate change (2T4), while women as before emphasizing the problem of increasing waste of disposable masks and gloves (2T9). In addition, women are more likely to say that confinement taught us a lesson to care for the environment (2T7), but they are also more likely to say that people quickly forget about the confinement (2T5).

Older people emphasize a lack of governmental support and that people face already many other problems (2T10 and 2T4), while younger generations talk about more responsible consumption while also noting increased waste due to the use of masks and gloves (2T1 and 2T9). People who perceive climate change as less important threat than COVID-19 express more the opinion that people prioritize economy and health (2T3) and talk less about the need to care for the planet (2T6).

## Putting the topics in context

Next, we try to better understand the affective valence of the generated topics, i.e. we assess the degree of optimism/pessimism of the expressed expectations. To this end, we draw on the closed-ended questions which preceded the open-ended ones and plot them against the two sets of topics. In Fig 3, the position of each topic on the X- and Y-axes reflects the weighted average responses on the closed questions. By weighted we mean that the average response per topic is calculated taking as the weight of the prevalence of the topic in the corresponding response. This way we take into account responses containing two or more topics. Topics plotted below the indifference line of 3 (standing for the "neither nor" response option on the five-point Likert scale) on the axes tend to correspond to negative expectations of the respondents

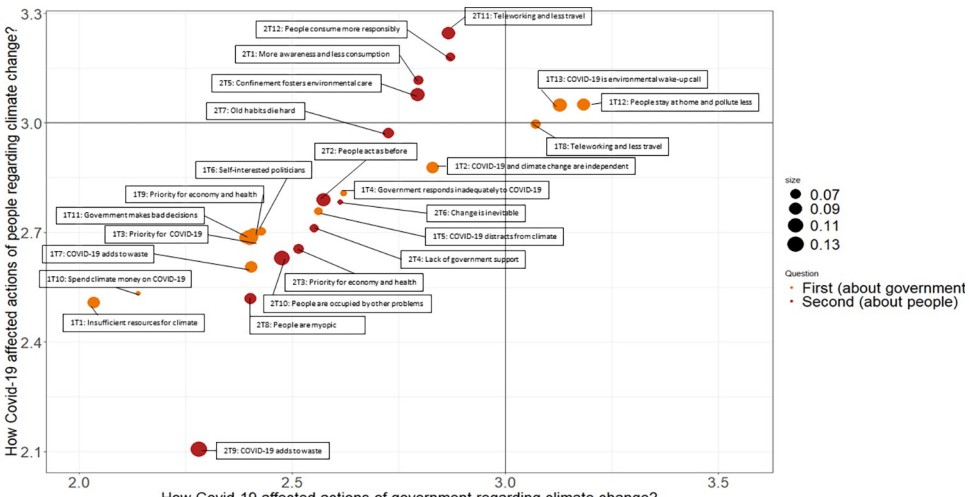

**Fig 3. Optimistic vs pessimistic views on actions of government and people with regard to climate for all topics generated.** Note: Orange color indicates topics from the first open question (about climate actions by the government), and red from the second one (about climate actions by the people). Size of the topic reflects its prevalence in the overall set of responses. Position of each topic on the X- and Y-axes is the weighted average response on the Likert scale from 1 (very negatively) to 5 (very positively) on the corresponding closed questions.

on how COVID-19 will affect action by the government or people with regard to climate change, while those above to positive ones. The two colors indicate which set of topics we are referring to (orange and red for the first and second open question, respectively), while the sizes of each bubble proxies the topic proportion in the whole set of responses.

The main question Fig 3 helps to answer is whether negative and positive expectations translate in similar discourses when applied to the government versus people. Since topics scoring high on one axis tend to do so also on the other axis, we see that optimistic expectations regarding governmental climate action tend to strongly correlate with those regarding people's climate action, and vice versa. Furthermore, few topics are positive, i.e. they do not surpass the threshold of 3 on the two closed questions. In fact, people are rather pessimistic in answering both closed questions, i.e. most answers are below the median response of 3 (the average responses are 2.60 and 2.77; see S3 Fig in S1 File). Still, our respondents are more optimistic concerning future action by the people than by the government. This is reflected in the lack of topics surpassing the threshold on the corresponding X-axis. These few topics (1T12 and 1T13) cover thoughts about reducing pollution due to confinement and the idea that COVID-19 can serve as a wake-up call. It is worth noting that people who expressed such beliefs tend to be also optimistic about future actions by the people. 1T8 also clearly surpasses the threshold for the corresponding closed question (with respect to the government) but is neutral with respect to people. In addition, four topics (2T1, 2T5, 2T11, 2T12) with respect to people's climate action managed to surpass only the threshold with regard to the actions by the people. A closer inspection of the content of these four topics indeed suggests positive views as to how COVID-19 affects action by the people (consume more responsibly, travel less or telework more).

The remaining 18 topics are below the threshold of 3 on both closed questions. Some of these topics are rather neutral and close to the threshold expecting that there will be no effect of COVID-19 and that people will quickly forget the confinement (1T2 and 2T7). In contrast, other topics are very pessimistic reaching an average score on closed questions of 2.1–2.5 and stress the lack of resources that can be spent on climate action in the future (1T1), while others hint at the waste people will generate in fighting the COVID-19 (2T9).

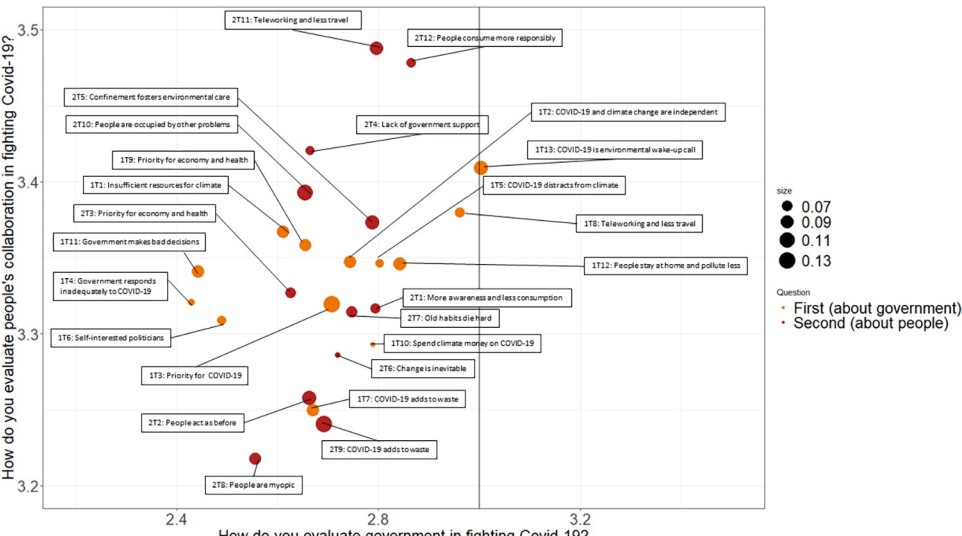

**Fig 4. Positive vs negative evaluation of on actions of government and people in fighting COVID-19 for all topics generated.** Note: Orange color indicates topics from the first open question (about climate actions by the government), and red from the second one (about climate actions by the people). Size of the topic reflects its prevalence in the overall set of responses. Position of each topic on the X- and Y-axes is the weighted average response on the Likert scale from 1 (very negatively) to 5 (very positively) on the corresponding closed questions.

In Fig 4 we repeat the exercise but now using two variables that capture people's collective efficacy beliefs. Specifically, they cover beliefs of how government and people work together to fight COVID-19. They are depicted on the X- and Y-axes, also on a scale from 1 to 5. As we know from S3 Fig in S1 File, respondents in our survey are much more positive in evaluating actions by the people than by the government (the average responses are 3.34 vs 2.71). Therefore, this time all topics surpass the threshold of 3 in evaluating people's actions but none of the topics surpasses the threshold in evaluating the actions by the government. The latter is an important result that can be partly explained by Spanish people tending to trust their peers much more than their government [31]. The topics that come closest to that threshold are again 1T8, 1T12 and 1T13 demonstrating that people who positively evaluate government in fighting against COVID-19 are also most optimistic regarding their future climate action.

## Discussion

It is worthwhile at this point to discuss our main results and compare them with findings provided recently in the literature. Little research has been undertaken on public expectations similar to our approach. A study by Lewandowsky et al. [32] for the UK and the US surveying people in a similar time period (May-July 2020) finds that respondents expect a 'return to normal' as more likely to happen than a sustainable post-pandemic future, which resonates with the relatively pessimistic expectations about future climate action identified in the present paper (e.g., topics 2T2 and 2T7 in Table 2 illustrate the expectation that old habits and routines will prevail). In addition, Lewandowsky et al. found that many people would like to maintain working from home and commute less, which is akin to our findings (topics 1T8, 1T12 and 2T11). This testifies that many results we obtained for Spain hold for other countries as well. Another study [33] conducted in the US on COVID-19 perceptions finds a substantial increase in economic anxiety, which is in line with several topics we find that prioritize government action on economic problems (1T1, 1T3, 1T5, 1T9, 1T10) and similarly citizen action (2T3 and 2T10).

Next, we link our findings to the results of several multi-country surveys. In 2021 the European Investment Bank conducted the fourth edition of the EIB Climate Survey to address the intersection between the climate crisis and the recovery from the COVID-19 crisis [34]. According to this survey, 83% of respondents from Spain (and 75% of EU citizens) believe they are more concerned about the climate emergency than their governments. A logical consequence of climate concern is personal climate action and/or support of climate policies [35]. This insight is in line with our result showing that people in Spain are much more positive about climate actions by their compatriots than by the government. Moreover, according to the EIB survey 81% of Spanish people (70% of EU citizens) are supporting stricter government measures to foster changes in people's behavior to mitigate climate change. These range from specific measures to replace short-distance flights by fast and low-polluting trains to systemic measures like carbon tax putting a price on goods and services proportional to their carbon footprint. In a companion study using the same survey, we show that after COVID-19, support of climate policies has increased [36]. This can be explained, among others, by the fact that people see COVID-19 and climate change as related threats, i.e. climate change is perceived as a contributing factor to the outbreak of COVID-19 or to the spread of infectious diseases more generally [37]. Some topics generated in the current study support this idea (e.g., 1T13). This suggests that policy makers could exploit this perceived link between the two problems to promote public support of climate policies and introduce stricter measures for climate change mitigation (see also [38]). It should be noted, however, that people also perceive various differences between climate change and the pandemic [39]. Incidentally, one of our topics (1T2) further suggests that some people see no connection between the pandemic and climate action. A recent survey conducted by the United Nations Development Programme (UNDP) and the University of Oxford [40] finds that clean transportation is one of the most popular climate policies. This echoes topic 2T11 in our study, which is about teleworking and clean transportation.

Overall, the findings of our study indicate that people have more negative than positive expectations about how the pandemic will affect climate action. This contrasts with research findings on actual changes of people's climate-relevant attitudes and behaviors. For example, one study for 16 countries shows that climate concern and policy support tend to have increased after COVID-19 [41]. This study, however, also indicates that income or job loss have negative effects on policy support, which resonates with our findings. As shown in Fig 2, people who were hit economically stronger by the COVID-19 pandemic express less the topic of environmental awareness (2T1). Nevertheless, another study shows that Spain was among the countries in which the amount of Twitter activity related to climate change remained relatively stable during the pandemic [42] reflecting that climate concern did not change much. Other research on actual behavioral changes due to the pandemic demonstrates that socio-political efficacy and in turn policy support has increased after COVID-19 [43]. Taken together, the rather negative expectations about citizen and government climate actions appear to contrast with actual observations. It is known that people underestimate others' support for climate policy [44] and the same seems to occur for expectations about future actions, meaning that people underestimate how many other citizens prefer a sustainable over status-quo future [32]. Providing information about actual attitudes, behaviors and expectations could help to countervail this gap, which in turn may facilitate climate action.

## Conclusion

Our study provides novel empirical insights on how citizens expect the COVID-19 crisis to affect future climate action by people and government. Using structural topic modelling, we

elicited thoughts on how the pandemic may influence governmental policies and citizen action against climate change. As pointed out already by Rosa et al. (2021), topic modelling serves as a powerful tool for summarizing content and sensemaking, hence clarifying citizens'voices and visions as an informative component for policy and foresight analysis. This is not trivial considering the increasing interest by governments to meaningfully streamline said visions into policy making and enable comparisons across countries. Combined with participatory data collection methods (including surveys), it presents an effective (in terms of insights) and efficient (in terms of time to process the data) tool to inform policy makers about citizens' views and expectations.

In general, there is some overlap between both sets of topics, which indicates similar expectations about the roles of government and people. The majority of identified topics regarding governmental action reflect negative perceptions. In general, they relate to reduced attention given to climate change, to budgetary constraints due to COVID-19 and the associated economic crisis, as well as to an increase of waste due to use of disposable protective measures like masks and gloves in the fight against COVID-19. A small number of people see little to no connection between COVID-19 and climate action. Only two topics covering about 15% of responses are of a more positive nature. They consider COVID-19 as an environmental wake-up call or point at positive changes in consumption habits and telework.

Out of twelve topics formed with respect to people's action, four can be seen as positive covering around 31% of responses, such as higher environmental awareness and more responsible consumption. Nevertheless, most respondents still express negative views, suggesting for example that people have already too many other problems to be concerned about climate action or will shift quickly back to old routines once the pandemic is over. Through measuring positive and negative expectations for each of the identified topics, we obtained several additional insights. First, expectations of future climate actions by the government and the people tend to be strongly correlated in their degree of optimism/pessimism. This is particularly true for those respondents who expressed positive views regarding future climate actions by the government, while those who expect positive changes in people's action can still be rather critical in their expectations about the government. Second, positive topics regarding future climate actions tend to be expressed by respondents who are younger, male, better educated, perceive climate change more as a threat, or had an overall more positive experience with COVID-19 confinement. Finally, we find respondents to be rather positive about how other people have participated in fighting COVID-19 so far, but more negative in how their government has fared in this regard. Interestingly, people who are most optimistic regarding future climate action by the government also evaluate the government positively with respect to fighting COVID-19.

These citizen views could be partly explained if one looks at the context in which they were created. Recent experiences with the global economic crisis and high unemployment rates in Spain may explain skepticism about the financial resources the government can direct on tackling climate change. Low levels of trust on politicians and fairly little coverage of the problem of climate change in Spanish media make the negative expectations in many topics more understandable. For policy makers such views can serve as a valuable source of information on what problems people see as critical, and what ways forward–such as teleworking or responsible consumption–they propose.

In summary, our study can be considered as a snapshot of public expectations, regarding future climate action by government and people affected by COVID-19, against which future developments can be compared. As the COVID-19 pandemic is still ongoing, new topics will and have already emerged, such as 'green' stimulus plans in response to the economic crisis. The present study can serve as a basis for comparing public perceptions over time and across

geographical contexts with different experiences than the one examined here, as well as for building and testing policy acceptance scenarios.

## Supporting information

**S1 File.**
(DOCX)

## Author Contributions

**Conceptualization:** Ivan Savin, Stefan Drews, Jeroen van den Bergh, Sergio Villamayor-Tomas.

**Data curation:** Ivan Savin.

**Formal analysis:** Ivan Savin.

**Writing – original draft:** Ivan Savin, Stefan Drews, Jeroen van den Bergh, Sergio Villamayor-Tomas.

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
