## [Decision Letter · Decision Letter 0]

28 Feb 2022

PONE-D-22-02316Public expectations about the impact of COVID-19 on climate action by citizens and governmentPLOS ONE

Dear Dr. Savin,

Thank you for submitting your manuscript to PLOS ONE. After careful consideration, we feel that it has merit but does not fully meet PLOS ONE’s publication criteria as it currently stands. Therefore, we invite you to submit a revised version of the manuscript that addresses the points raised during the review process.

We look forward to receiving your revised manuscript.

Kind regards,

Ghaffar Ali, PhD

Academic Editor

PLOS ONE

Journal Requirements:

[This work was funded by an ERC Advanced Grant from the European Research Council (ERC) under the European Union’s Horizon 2020 Research and Innovation Programme [grant agreement n° 741087] and by a María de Maeztu Program for Units of Excellence awarded to ICTA-UAB by the Spanish Ministry of Science and Innovation. I.S. acknowledges financial support from the Russian Science Foundation [RSF grant number 19-18-00262]. S.V.-T. acknowledges financial support from the Spanish Department of Science and Innovation [RYC-2017-22782].]

[This work was funded by an ERC Advanced Grant from the European Research Council (ERC) under the European Union’s Horizon 2020 Research and Innovation Programme [grant agreement n° 741087] and by a María de Maeztu Program for Units of Excellence awarded to ICTA-UAB by the Spanish Ministry of Science and Innovation. I.S. acknowledges financial support from the Russian Science Foundation [RSF grant number 19-18-00262]. S.V.-T. acknowledges financial support from the Spanish Department of Science and Innovation [RYC-2017-22782]]

Reviewers' comments:

Reviewer's Responses to Questions

**Comments to the Author**

1. Is the manuscript technically sound, and do the data support the conclusions?

Reviewer #1: Partly

Reviewer #2: Partly

2. Has the statistical analysis been performed appropriately and rigorously? 

Reviewer #1: Yes

Reviewer #2: Yes

3. Have the authors made all data underlying the findings in their manuscript fully available?

Reviewer #1: Yes

Reviewer #2: Yes

4. Is the manuscript presented in an intelligible fashion and written in standard English?

Reviewer #1: Yes

Reviewer #2: Yes

5. Review Comments to the Author

Reviewer #1: The article addresses the critical issue of public expectations about the impact of COVID-19 on climate action by citizens and government. The article is interesting and is of high quality. However, I have some below-mentioned concerns.

1. In the section "2.1 Data collection" the authors stated that "In this study we draw on two open-ended questions from an online survey conducted in June-July 2020 in Spain." The author should explicitly mention what kind of online platforms were used to collect responses, e.g., social media (Facebook, LinkedIn, Twitter, WhatsApp, etc.) and/or email? Although the authors have mentioned, "The survey was conducted in Spanish language by a professional panel survey company "Netquest. " But I still believe the individual data collection sources should be indicated.

2. What techniques were employed to cope with the sample bias?

3. The authors used an online survey in this study, which is unquestionably a more convenient, robust, and cost-effective survey method. However, the authors should acknowledge in the paper that these types of surveys have limitations. For example, authors can mention the survey's inability to reach difficult populations such as the elderly and those with no educational background.

4. Furthermore, a discussion section should be included that interprets and describes the significance of your findings in light of what was already known about the research problem being investigated, as well as to explain any new understanding or fresh insights about the problem after you've taken the findings (any novel or unexpected results) into consideration. In my opinion, the inclusion of a discussion section can add to the quality of the findings of your article.

5. Finally, there are some typo errors that should be corrected in the revised version.

Reviewer #2: In this paper “Public expectations about the impact of COVID-19 on climate action by citizens and government” authors presented the Spanish citizen expectation about the COVID-19 impact on climate action by the public and government. This paper is a good attempt at understanding public opinion regarding climate action and the pandemic impact on climate impact. There are several gaps that still need to be addressed in this paper before publication.

Details comments have been added to the separate document in the attachments. The authors should read the attached document. Decision - Major Revision.

6. PLOS authors have the option to publish the peer review history of their article (what does this mean?). If published, this will include your full peer review and any attached files.

Reviewer #1: No

Reviewer #2: No

---

## [Author Response · Author response to Decision Letter 0]

16 Mar 2022

Please read the attached response to editor and reviewers with detailed responses

---

## [Decision Letter · Decision Letter 1]

31 Mar 2022

Public expectations about the impact of COVID-19 on climate action by citizens and government

PONE-D-22-02316R1

Dear Dr. Savin,

We’re pleased to inform you that your manuscript has been judged scientifically suitable for publication and will be formally accepted for publication once it meets all outstanding technical requirements.

Kind regards,

Ghaffar Ali, PhD

Academic Editor

PLOS ONE

Additional Editor Comments (optional):

Reviewers' comments:

Reviewer's Responses to Questions

**Comments to the Author**

1. If the authors have adequately addressed your comments raised in a previous round of review and you feel that this manuscript is now acceptable for publication, you may indicate that here to bypass the “Comments to the Author” section, enter your conflict of interest statement in the “Confidential to Editor” section, and submit your "Accept" recommendation.

Reviewer #1: All comments have been addressed

Reviewer #2: All comments have been addressed

2. Is the manuscript technically sound, and do the data support the conclusions?

Reviewer #1: Yes

Reviewer #2: Yes

3. Has the statistical analysis been performed appropriately and rigorously? 

Reviewer #1: Yes

Reviewer #2: Yes

4. Have the authors made all data underlying the findings in their manuscript fully available?

Reviewer #1: Yes

Reviewer #2: Yes

5. Is the manuscript presented in an intelligible fashion and written in standard English?

Reviewer #1: Yes

Reviewer #2: Yes

6. Review Comments to the Author

Reviewer #1: Thanks for incorporating my suggestions. I believe the article in its current form can be published by PLOS ONE.

Reviewer #2: Authors have addressed the comments i have made. I don't have any further comments on this paper. Just make sure final submission have formatting according to the journal requirement.

7. PLOS authors have the option to publish the peer review history of their article (what does this mean?). If published, this will include your full peer review and any attached files.

Reviewer #1: No

Reviewer #2: No

---

## [Editor Report · Acceptance letter]

17 May 2022

PONE-D-22-02316R1 

Public expectations about the impact of COVID-19 on climate action by citizens and government 

Dear Dr. Savin:

I'm pleased to inform you that your manuscript has been deemed suitable for publication in PLOS ONE. Congratulations! Your manuscript is now with our production department. 

Kind regards, 

on behalf of

Prof. Ghaffar Ali 

Academic Editor

PLOS ONE